# Synthesis Characterization and Highly Protective Efficiency of Tetraglycidyloxy Pentanal Epoxy Prepolymer as a Potential Corrosion Inhibitor for Mild Steel in 1 M HCl Medium

**DOI:** 10.3390/polym14153100

**Published:** 2022-07-30

**Authors:** Rachid Hsissou, Redouane Lachhab, Anouar El Magri, Siham Echihi, Hamid Reza Vanaei, Mouhsine Galai, Mohamed Ebn Touhami, Mohamed Rafik

**Affiliations:** 1Laboratory of Organic Chemistry, Catalysis and Environment, Department of Chemistry, Faculty of Sciences, Ibn Tofail University, BP 242, Kenitra 14000, Morocco; lrafik@yahoo.com; 2Laboratory of Advanced Materials and Process Engineering, Faculty of Sciences, Ibn Tofail University, BP 242, Kenitra 14000, Morocco; lachhabred1@gmail.com (R.L.); galaimouhsine@gmail.com (M.G.); mohamed.ebntouhami@uit.ac.ma (M.E.T.); 3Euromed Polytechnic School, Euromed Research Center, Euromed University of Fes, Route de Meknès (Rond point Bensouda), Fès 30000, Morocco; 4Laboratory of Materials Engineering for the Environment and Natural Resources, Faculty of Sciences and Techniques, University Moulay Ismail of Meknes, BP 509 Boutalamine, Errachidia 52000, Morocco; echihisiham@gmail.com; 5Arts et Métiers Institute of Technology, CNAM, LIFSE, HESAM University, F-75013 Paris, France

**Keywords:** synthesis, epoxy prepolymer TGP, FTIR/NMR characterization, anticorrosive protection, PDP/EIS measurements, SEM/EDS/CA analyses, Langmuir adsorption

## Abstract

Anticorrosive protection efficiency of novel tetrafunctional epoxy prepolymer, namely 2,3,4,5-tetraglycidyloxy pentanal (TGP), for mild steel in 1 M HCl medium was assessed through potentiodynamic polarization (PDP), electrochemical impedance spectroscopy (EIS), scanning electron microscope (SEM), energy dispersive X-ray spectroscopy (EDS), contact angle (CA), adsorption isotherm model, temperature effect and thermodynamic parameters. The synthesized TGP was characterized and confirmed by Fourier transform infrared (FTIR) spectroscopy and nuclear magnetic resonance (NMR). The inhibitory efficiencies found at lower concentration of the prepolymer TGP were85% (PDP) and 87.17% (EIS). PDP measurement illustrated that the TGP behaved as a mixed-type inhibitor in the realized solution. SEM and EDS analysis showeda significant decrease in the corrosion of the MS surface in the presence of the inhibitory prepolymer compared with the blank (1 M HCl). Langmuir adsorption isotherm is the most acceptable modelto describe the TGP epoxy prepolymer on the MS area.

## 1. Introduction

Mild steel surfacesarewidely employed and investigated in large civil engineering projects because of theiradvantages of availability and excellent tensile and mechanical strength [1,2]. The introduction of organic inhibitors such as quinoxaline, quinoline, polymer composites and epoxy prepolymers has become one of the most effective approaches for retarding the corrosion of mild steel in different aggressive solutions [3,4,5]. Multifunctional epoxy prepolymer molecules possessing oxygen, sulfur and nitrogen heteroatoms and electronegative polar groups, as well as aldehydes, carboxyl, amino, oxirane groups have been used to display excellent protective abilities ascorrosion inhibitors by forming a chemical bond with the metallic surface [6,7,8,9]. Epoxy prepolymer molecules have π-electrons and unsaturated donating and are able to accept electrons in their low-energy empty orbitals [10,11]. Epoxy prepolymer compounds have been investigated as excellent corrosion inhibitors because of their ability to adsorb readily to ametallic surface [12,13,14]. This class of epoxy prepolymer molecules has been investigated as corrosion inhibitors because of its synthesis by the condensation reaction of epichlorohydrin with compounds having mobile hydrogen in the presence of sodium hydroxide (NaOH) as the basis [15,16,17]. The presence of an oxirane group, carbonyl compounds and oxygen heteroatoms enables it to cooperate with an iron surface, facilitating the adsorption of epoxy prepolymer on the metallic area and acting as a corrosion inhibitor [18,19,20,21]. Damej et al. studied N,N-1-tri(oxiran-2-ylmethoxy)-5-((oxiran-2-ylmethoxy)thio)-1H-1,2,4-triazole-3-amine as a potential corrosion inhibitor for steel in 1 M HCl solution and found ahigh inhibitory efficiency of 92% at lower concentration (1mM) [6].

In this work, the novel tetrafunctional epoxy prepolymer, namely,2,3,4,5-tetraglycidyloxy pentanal (TGP), was synthesized by the condensation reaction of arabinose with epichlorohydrin in the presence of triethylamine as the basis. The synthesized TGP was characterized and confirmed by Fourier transform infrared (FTIR) spectroscopy and nuclear magnetic resonance (NMR). Additionally, TGP was used and investigated as a potential corrosion inhibitor for mild steel in 1 M HCl solution by potentiodynamic polarization technique, electrochemical impedance spectroscopy measurements, scanning electron microscope (SEM) analysis, energy dispersive X-ray spectroscopy (EDS) study and contact angle (CA) technique [22]. The last two decades have witnessed great progress in the prediction of the properties and chemical reactivity of different materials, despite their complexities, due to the development and advancement of numerical simulation systems. In addition, the adsorption isotherm model, temperature effect and thermodynamic parameters were investigated and thoroughly discussed [23,24,25].

## 2. Materials and Methods

### 2.1. Synthesis of 2,3,4,5-Tetraglycidyloxy Pentanal (TGP)

All chemical products used in this experimental section, such as arabinose (99%), epichlorohydrin (99%), methanol (99%) and triethylamine (99.5%), were purchased from Sigma Aldrich, Germany. All chemical products were employed without other purification. Novel tetrafunctional epoxy prepolymer, namely, 2,3,4,5-tetraglycidyloxy pentanal (TGP), was elaborated and developed by condensation reaction in two steps according to the procedure reported in the literature [15,16,26,27]. In a balloon of 100 mL fitted with refrigerant, 4.73 × 10^−3^ mol of the arabinose was dissociated in the methanol as solvent after we added 2.42 × 10^−2^ mol of epichlorohydrin under magnetic stirring at 70 °C for 6 h (Figure 1). Additionally, we condensed 2.54 × 10^−2^ mol of triethylamine as the basis for thereaction mixture under magnetic stirring at 40 °C for 4 h (Figure 1). Furthermore, methanol and triethylamine excess was removed with a rotary evaporator. Finally, we obtained a prepolymer of a brown color.

### 2.2. Tested Material

The chemical composition of mild steel (MS) substrates used in the experimental section were C(14.55%), Si(0.68%), S(0.41%), Cr(15.64%), Mn(1.94%), Fe(57.99%), Ni(7.92%) and Mo(0.86%). The treatment of the MS electrode usedhas been widely reported in the literature [1,2,22].

### 2.3. Characterization Techniques

All functional groups of novel tetrafunctional epoxy prepolymer, namely, the 2,3,4,5-tetraglycidyloxy pentanal (TGP) synthesized, were identified and characterized by Fourier transform infrared (FTIR) and nuclear magnetic resonance (NMR), respectively. FTIR spectra were obtained with a Bruker AVANCE (300 MHz) spectrometer at room temperature. The light beam passed through the 5 mg sample. The analysis wascarried out between 4000 and 400 cm^−1^, which corresponds to the energy range of the vibration of chemical bonds of organic molecules. Furthermore, NMR spectrometry was performed using a Bruker AVANCE (300 MHz) apparatus. The sample was dissolved in deuterated DMSO (DMSO-d6), and the chemical shifts wereexpressed in ppm. Additionally, the MS electrode area plunged in 1 M HCl medium, bothunprotected and protected with a lower concentration of TGP (10^−3^ M), was investigated by the scanning electron microscope (SEM) technique, energy dispersive X-ray (EDS) analysis and contact angle (CA) measurements. The SEM analysis was studied to make images of the MS electrode, both unprotected and protected with 10^−3^ M of TGP, using a JEOL-JSM-5500 microscope Type. Then, EDS was used to determine the elemental chemical composition. The conditions of the measurement were as follows: the acceleration voltage was 15.00 kV, the magnification wasx 3500, the live time was 30.00 s, the real time was 31.97 s, the dead time was 6.00% and the count rate was 19,403.00 CPS. In addition, the electrochemical study was carried out in a three-electrodes cell using a Potentiostat/Galvanostat/SP-200 biological apparatus. The work electrode was mild steel (MS), the counter electrode was platinum (Pt) and the reference electrode was a saturated calomel electrode (SCE). Open circuit potential (OCP) was realized for 30 min foreach assay after reaching the equilibrium state. The EIS study was carried out at OCP at afrequency range from 100 KHz to 10 mHz with 10 mV of signal amplitude. In addition, potentiodynamic polarization (PDP) was measured at 0.2 mV/s during potential range from −900 to −0.1 mV. Furthermore, the inhibition efficiencies for potentiodynamic polarization ƞ_PDP_ and electrochemical impedance spectroscopy ƞ_EIS_ were determined according to Equations (1) and (2):(1)$$\eta_{\mathrm{PDP}}\;(\%)=\frac{i_{\mathrm{corr}}^{0}{\;-\;i}_{\mathrm{corr}}}{i_{\mathrm{corr}}^{0}}\times 100$$
(2)$$\eta_{\mathrm{EIS}}\;(\%)=\frac{R_{\mathrm{ct}}{\;-\;R}_{\mathrm{ct}}^{0}}{R_{\mathrm{ct}}}\times 100$$

$R_{\mathrm{ct}}^{0}$, $R_{\mathrm{ct}}$, $i_{\mathrm{corr}}^{0}$ and $i_{\mathrm{corr}}$ represent the charge transfer resistance unprotected, charge transfer resistance protected with varying concentrations of TGP, unprotected corrosion current densities and protected corrosion current densities with different concentrations of TGP, respectively.

## 3. Results

### 3.1. FTIR Characterization

The novel tetrafunctional epoxy prepolymer, namely, the 2,3,4,5-tetraglycidyloxy pentanal (TGP) synthesized, was recorded by utilizing Fourier transform infrared (FTIR) spectroscopy (BRUKER type) to determine and characterize the varying functional groups. The FTIR spectrum analysis of TGP is illustrated in Figure 1.

The absorption band in the FTIR spectrum of TGP appeared at 3324 cm^−1^, which is attributed to the stretching vibrations of hydroxyl function (-OH) of an unclosed epoxy group [15]. The absorption band situated at 2973 cm^−1^ correspondsto aliphatic methylene (CH_2_) [15,16]. Furthermore, the absorption band assigned at 2845 cm^−1^ correspondsto C-H of aldehyde function [16]. Then, the absorption band located at 1738 cm^−1^ correspondsto aldehyde function (C=O) [27]. Additionally, the absorption bands at 1432, 1396 and 1366 cm^−1^ correspond to the stretching vibrations of aliphatic C-H [28]. Furthermore, the absorption bands which appeared at 1090, 1050 and 1005 cm^−1^ can be assigned to the asymmetric stretching vibration of C-O-C aliphatic ether function [15]. In addition, the presence of the epoxy group is displayed by the characteristic absorption bands at 902 and 853 cm^−1^, which areattributed to stretching of C-O-C and C-O of oxirane group [15,27].

### 3.2. NMR Characterization

The synthesized 2,3,4,5-tetraglycidyloxy pentanal (TGP) was characterized and confirmed using ^13^C NMR spectroscopy to determine the carbon atoms. The NMR spectrum of TGP is illustrated in Figure 2.

The attribution of varying chemical shifts of the synthesized epoxy prepolymer is as follows:

^13^C RMN (ppm): 47 (s, CH_2_ of oxirane group), 52 (s, CH of oxirane group), 55 (s, CH_2_ related to oxirane and O-CH_2_) and 72 (s, CH_2_ linked to O and aliphatic CH).

### 3.3. PDP Analysis

To know and understand the kinetics of the electrochemical reactions occurring at the active centers of the MS electrode surface, both unprotected and protected by varying concentrations of the studied epoxy prepolymer in 1 M HCl solution at 298 K, potentiodynamicpolarization curves were used and investigated as shown in Figure 3 and Table 1. Theresults illustrated in Table 1 show that the corrosion current density (i_corr_) diminished from 983.0 μA cm^−2^ for unprotected to 147.42 μA cm^−2^ for protected within the highest concentration of studied epoxy prepolymer (10^−3^ M) [3,29]. Additionally, within the lowest concentration (10^−6^ M), the inhibitory efficiency reached avalue of 76.88% and gradually increased within the highest concentration (10^−3^ M) of the TGP epoxy prepolymer, reaching a highest value of 85.0 % [16]. These results showedthat the TGP epoxy prepolymer at low concentration could form a protective layer on the MS area by reducing the number of active centers and be used and investigated as a good inhibitor. Furthermore, in this experimental part, the difference between the potential (ΔE_corr_) of the unprotected and protected with varying concentrations of TGP (from 21 to 45 mV) was less than 85 mV, showing that the investigated epoxy prepolymer actsas a mixed-type inhibitor [30,31]. These results revealed that the TGP epoxy prepolymer was able to stop substrate electrode dissolution at the anode as well as hydrogen gas evolution at the cathode. According to the data displayed in Table 1, careful analysis of the anodic and cathodic Tafel slopes suggests that, with the addition of the TGP prepolymer, both slopes were changed compared to the 1 M HCl solution alone [32]. Additionally, the reaction mechanisms, both anodic and cathodic, for the unprotected and protected by different concentrations of the epoxy prepolymer were confirmed to be constant as the shape of the polarization curves remained the same, showingthat the epoxy prepolymer synthesized adsorbed on the metallic area through blocking the active centers [6,20].

### 3.4. EIS Analysis

To understand and evaluate the corrosion inhibition of MS electrode surfaces, the elaborated epoxy prepolymer was used and investigated. Figure 4 displays the Nyquist curves for the anticorrosion protection property of the epoxy prepolymer inhibitor in an acidic solution, bothunprotected and protected, with varying concentrations of the studied inhibitor. Additionally, careful examination of the Nyquist curves suggests that the corrosion inhibition mechanism was the same for the unprotected and protected experiments owing to the semicircle loops [7,33]. As indicated in Figure 4, judging from the significant increases in the sizes of the semicircles and the charge transfer resistance in the different concentrations of epoxy prepolymer, one could conclude that the TGP synthesized was able to adsorb on the metallic substrates, forming a barrier and restricting the metal dissolution [34,35]. Then, EIS results were subsequently fitted from an equivalent circuit comprising the constant phase element (CPE), the solution resistance (R_s_) and the charge transfer resistance (R_ct_), as displayed in Figure 4. The EIS parameters are listed in Table 2, and the analysis of the results reflected an important increase in the values of charge transfer resistance from 34.81 to 210.2 Ω cm^2^ for 1 M HCl alone and with the 10^−3^ M of TGP epoxy prepolymer, respectively, showing the highest charge transfer resistance through the tested inhibitor at the MS/electrolyte interface [17]. In addition, the highest charge transfer resistancevalues, as shown in Table 2, confirmed the inhibitory efficiency of the epoxy prepolymer in the corrosion inhibition of metallic surfaces in 1 M HCl solution. Furthermore, the phase element constant values are revealed in the different concentrations of epoxy prepolymer in the 1 M HCl solution, resulting in the formation of a covering layer on the MS surface through epoxy prepolymer, and the increases in the thickness of protective layer as more H_2_O molecules were replacedbythe epoxy prepolymer at highest concentrations [36,37,38]. As seenin Table 2, the n_dl_ values were investigated to classify the CPE as well as revealing the nature of the metallic substrates’ electrode area. According to data displayed in Table 2, the n_dl_ values of both the unprotected and protected MSby different concentrations of epoxy prepolymer appeared to be lowerand close to 1, suggesting the pseudocapacitive nature of the CPE. These results indicate that the TGP epoxy prepolymer is a good inhibitor compared to other macromolecules used as corrosion inhibitors [39,40].

### 3.5. SEM/EDS Characterization

To know and understand the adsorption mechanism between the MS electrode surface and the synthesized inhibitory prepolymer, SEM and EDS techniques were investigated and discussed. Figure 5 illustrates the SEM images of the MS electrode plunged in 1 M HCl solution for unprotected (a) and protected by lower concentration (10^−3^ M) of used inhibitory resin (b). In fact, SEM images show that the surface of the MS electrode hadbecome rough due to the deposition of a protective layer [22]. Figure 6 illustrates the X-ray energy dispersive spectroscopy (EDS) analysis of the MS surface, for both unprotected and that protected with a lower concentration of the used inhibitor, in 1 M HCl medium. According to the results displayed in Figure 6, it can be seen that the iron content of the MS surface increases from 66.03 to 66.16%, and the Cr, Mn, Ni and Mo elemental peaks disappear with added TGP inhibitory prepolymer at a lower concentration. Furthermore, elemental oxygen content peaks appeared, and the carbon content increased in the presence of 10^−3^ M of the studied inhibitor due to the oxygen heteroatoms and carbon present in the molecule of TGP prepolymer synthesized, resulting inthe inhibitory resin reducing the density of active centers on the MS area by forming the protective layer and retarding corrosion inhibition [40].

### 3.6. Contact Angle Characterization

The layer that forms on the MS electrode surface and its hydrophilic or hydrophobic characteristics can be understood through contact angle analysis of MS surfaces that are both unprotected and protected by 10^−3^ M of TGP prepolymer after being in contact with a 1 M HCl solution (Figure 7). The unprotected MS area in diiodomethane, water and formamide had contact angles of 12.5, 30.2 and 19.6°, respectively. These data can be explained by the hydrophilicity characteristic (favors water). The formation of some polar inorganic corrosion products is responsible for the hydrophilic phenomena. However, the contact angles of the MS surface protected by 10^−3^ M of TGP in diiodomethane, water and formamide were 65.4, 89.9 and 62°, respectively. These data indicate that the MS surface protected by 10^−3^ M of TGP has a hydrophobic character (does not favor water). This result confirms the formation of a hydrophobic layer on the MS electrode area. By comparing between the unprotected MS and that protected by 10^−3^ M of TGP, we can conclude that the MS electrode surface protected with 10^−3^ M of TGP inhibitor has a more hydrophobic character than the unprotected MS, leading to more prevention of corrosion attacking the steel surface. Additionally, the total energy of the MS surfaces, both unprotected and that protected with 10^−3^ M of TGP, was 54.7 and 28.6 mJ/m^2^, respectively. The results in Figure 7 confirm that ahydrophilic layer has formed on the MS area [22].

### 3.7. Adsorption Study

To know the adsorption mechanism between the TGP epoxy prepolymer and a mild steel electrode surface in 1 M HCl medium, experimental results obtained from both PDP and EIS measurements were calculated usingthe various adsorption isotherm equations and the best fit, which, in this case, was the Langmuir adsorption isotherm model, as shown in Figure 8. The results from PDP and EIS analysis indicate that the TGP prepolymer synthesized was adsorbed on the metallic substrates by forming a protective layer that protects the mild steel against corrosion [40]. The obtained parameters from the studied Langmuir adsorption for theinhibitory resin are illustrated in Table 3. According to the results obtained from the two techniques (PDP and EIS), the coefficient values (R^2^) are 1 and 0.9999 for PDP and EIS, respectively, which conformed well to the Langmuir adsorption isotherm. These results show that the inhibitory prepolymer can be adsorbed on the mild steel area accordingto the best model that can be used to interpret this particular phenomenon. Adsorption Gibbs energy variation values (ΔG_ads_) give more information about the adsorption mode of the inhibitor and the MS area. Additionally, the ΔG_ads_ values werehigher than −40 KJ mol^−1^ and less than −20 kJ mol^−1^, indicating the chemisorption and physisorption modes, respectively [22]. The Langmuir adsorption isotherm model and the adsorption Gibbs energy variation could be calculated according to Equations (3) and (4):(3)$$\frac{C_{\mathrm{inh}}}{\theta }=\frac{1}{K_{\mathrm{ads}}}+C_{\mathrm{inh}}$$
(4)$${\Delta G}^{{}^{\circ}}{}_{\mathrm{ads}}=-R\times T\times \mathrm{Ln}(55.5\times K_{\mathrm{ads}})$$

C_inh_, θ, K_ads_, R and T are the inhibitory concentration, the surface coverage degree, the adsorption equilibrium constant, the perfect gas constant and the used temperature, respectively.

As the data displayed in Table 3 show, the ΔG_ads_ values for PDP and EIS analyses were 42.13 and 43.74, respectively, suggesting that the epoxy prepolymer inhibitory adsorbed on the MS area by chemisorption mode. Furthermore, the chemisorption model of the TGP prepolymer could be due to the bond types between iron ions and the bond double of carbonyl compound (C=O) and free electron pairs of oxygen heteroatoms present in the synthesized prepolymer.

### 3.8. Temperature Effect and Thermodynamic Parameters

From the experimental data (PDP and EIS measurements), it was established that the used inhibitoryepoxyresin was adsorbed on the MS area by forming aprotective layer that protects the metal against corrosion. So, the inhibitory efficiency needs to be used and investigated at different temperatures (298, 308, 318 and 328K) using the PDP technique. Figure 9 illustrates the PDP plots forthe MS electrode both unprotected and protected by 10^−3^ M of the studied inhibitor. As the data illustrate in Table 4, the corrosion current density of the MS electrode, bothunprotected and protected by 10^−3^ M, increases with increasing temperatures and its inhibitory efficiency decreased, showingthat the increases in temperature accelerated the corrosion of the MS surface in the 1 M HCl medium, and it was found that the inhibitory efficiency of the MS electrode in 1 M HCl solution after adding thelowest concentration decreases from 85 to 81.62% for 298 and 328K, respectively [22]. This behavior can be explained by the weakening bond of the inhibitory TGP prepolymer, resulting in areduction inthe coverage surface and its inhibitory efficiency decreasing.

The Arrhenius model was usedto determine the activation energy of the MS electrode surface according to Equation (5). Additionally, the enthalpy and entropy activation were calculated by the following Equation (6):(5)$$i_{\mathrm{corr}}=\mathrm{Aexp}\left(-\frac{E_{a}}{RT}\right)$$
(6)$$i_{\mathrm{corr}}=\frac{RT}{hN}\exp\left(\frac{\Delta S_{a}}{T}\right)\exp\left(-\frac{\Delta H_{a}}{RT}\right)$$

N, T, E_a_, R, H, K, ΔH_a_ and ΔS_a_ are Avogadro’s number, the absolute temperature, the activation energy, the perfect gas constant, Planck’s constant, the pre-exponential factor, the activation enthalpy and the activation entropy, respectively.

Figure 10 illustrates the Log(i_corr_) as a function of 1000/T and Log (i_corr_/T) as a function of 1000/T curves. As the data display in Table 4, Ea reflects the minimum energy required for an effective collision between inhibitory resin and MS surface. According to the results listed in Table 4, the Ea value significantly increases, indicating that the corrosion of the MS surface increases in the presence of the inhibitory resin due to the TGP prepolymer being adsorbed on the metallic surface, which occupiesthe active centers ofthe reaction and inhibits the corrosion reaction. Furthermore, the activation energy of the synthesized resin is higher than that of the 1 M HCl alone, reflecting that the investigated molecule has good anticorrosive protection. Furthermore, the activation enthalpy positively indicates that the dissolution of the MS surface added to corrosion inhibition is a heat absorption process. The activation entropy negatively indicates the activation complex is an interaction phase and not a dissociation phase in the rate-determining phase. According to the presented data in Table 4, with the inhibitoryprepolymer, the ΔS_a_ value is higher than that of the 1 M HCl only, resulting in the formation of aprotective layer on the MS surface.

## 4. Conclusions

The corrosion inhibition of a tetrafunctional epoxy prepolymer (TGP) has been studied via electrochemical measurements, surface analyses and calculation studies. The results from the PDP data suggest that the TGP adsorbed strongly onto the MS surface, behaving as a mixed-type corrosion inhibitor. The Langmuir adsorption isotherm and slope indicate monolayer adsorption with R^2^ value approaching unity. PDP and EIS have high inhibitory efficiencies at lower concentration of the TGP. The Gibbs free energy values for the adsorption of the TGP prepolymer indicates chemisorption. SEM and EDS analysis reveal the potential ability of TGP in the 1 M HCl solution. Additionally, the adsorption isotherm obeys the Langmuir model.

## Data Availability

The data that support the findings of this study are available from the corresponding author upon reasonable request.

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
