# Peer review of "Synthesis Characterization and Highly Protective Efficiency of Tetraglycidyloxy Pentanal Epoxy Prepolymer as a Potential Corrosion Inhibitor for Mild Steel in 1 M HCl Medium"

_polymers, 2022, doi:10.3390/polym14153100_

Round 1

Reviewer 1 Report

Comments to the authors

In this manuscript, the authors synthesized a new tetrafunctional epoxy prepolymer according to the procedure reported in the literature, evaluated its corrosion protection effect on mild steel in 1 M HCl medium, and determined its adsorption isotherm model. The innovation of this study is moderate, and the subject addressed in this article is worthy of investigation. But there are some errors and distortions in the manuscript. This article could be accepted after minor revision, especially pay attention to the following questions:

1. There are spelling and grammatical errors in this article, please correct them carefully. For example, there is an obvious error on line 54.

2. For the characterization results in this paper (such as contact angle characterization), it is suggested to supplement more analysis and discussion, rather than simply describe the data and phenomena.

3. In Figure 1, the peak at 2400 cm-1 is not the C-H absorption peak of aldehyde, but may be the absorption peak of carbon dioxide. The C-H absorption peak of aldehyde should appear at 2850~2710 cm-1.

4. The expression "the MS SPECIMES shows the protective layer and smooth with fewer cracks and holes..." mentioned in line 205 on page 6 is not accurate. In fact, SEM images show that the surface of MS electrode has become rough due to the deposition of a protective layer.

Author Response

Reviewer #2: In this manuscript, the authors synthesized a new tetrafunctional epoxy prepolymer according to the procedure reported in the literature, evaluated its corrosion protection effect on mild steel in 1 M HCl medium, and determined its adsorption isotherm model. The innovation of this study is moderate, and the subject addressed in this article is worthy of investigation. But there are some errors and distortions in the manuscript. This article could be accepted after minor revision, especially pay attention to the following questions:

Comment 1: There are spelling and grammatical errors in this article, please correct them carefully. For example, there is an obvious error on line 54.

Answer : Thanks for your kind and suggestive comments. All spelling and grammatical errors have been revised (see manuscript)

Comment 2: For the characterization results in this paper (such as contact angle characterization), it is suggested to supplement more analysis and discussion, rather than simply describe the data and phenomena.

Answer : Thank you very much for your professional question. The contact angle characterization has been revised and improved (see manuscript).

Comment 3: In Figure 1, the peak at 2400 cm-1 is not the C-H absorption peak of aldehyde, but may be the absorption peak of carbon dioxide. The C-H absorption peak of aldehyde should appear at 2850~2710 cm-1.

Answer : Dear reviewer, we think that you raised a very good question. The C-H absorption peak of aldehyde has been assigned at 2845.

Comment 4: The expression "the MS SPECIMES shows the protective layer and smooth with fewer cracks and holes..." mentioned in line 205 on page 6 is not accurate. In fact, SEM images show that the surface of MS electrode has become rough due to the deposition of a protective layer.

Answer : Dear reviewer, the sentence "the MS SPECIMES shows the protective layer and smooth with fewer cracks and holes..." has been revised as "In fact, SEM images show that the surface of MS electrode has become rough due to the deposition of a protective layer".

The authors would like to thank you for your time and effort. Thanks to these constructive comments,

Reviewer 2 Report

The manuscript describes the epoxy prepolymer as a potential corrosion inhibitor for mild steel under the condition of 1M HCl solution. The manuscript is well-organized and provides some information for audiences. However, there are some minor points to be addressed as below.

1. In Scheme 1, methanol/70C/8h was displayed. However, in the main text, 6h was written. Which one is correct?

2. The hydroxyl moieties of arabinose were reacted with epichlorohydrin. I was wondering if the methanol as a solvent was reacted with epichlorohydrin as well?

3. Details for characterization techniques should be provided such as temperature, conditions, gas flow rate, heating rate, etc.

4. Proper references for the absorption bands at 2973, 2400, 1738, 1432, 1396, and 1366 cm-1 should be cited in the main text.

5. Only meaningful values in the NMR graph should be displayed and discussed.

6. What is the driving force or mechanism for the epoxy prepolymer investigated in this study acting as an effective corrosion inhibitor for mild steel?

7. Plenty of typos and grammar errors:

enables the to cooperate??

All used chemical products used in this -> Remove "used"

different carbons -> carbon atoms.

prepolymer was used and investigate -> investigated

a important -> an

epoxy prepolymer, which resulting -> Remove "which"

Author Response

Reviewer #1: The manuscript describes the epoxy prepolymer as a potential corrosion inhibitor for mild steel under the condition of 1M HCl solution. The manuscript is well-organized and provides some information for audiences. However, there are some minor points to be addressed as below.

Comment 1: In Scheme 1, methanol/70C/8h was displayed. However, in the main text, 6h was written. Which one is correct?

Answer : Thanks for your kind and suggestive comments, "methanol/70C/8h" in scheme 1 has been revised as " methanol/70C/6h".

Comment 2: The hydroxyl moieties of arabinose were reacted with epichlorohydrin. I was wondering if the methanol as a solvent was reacted with epichlorohydrin as well?

Answer : Thank you very much for your professional question. Methanol does not react with epichlorohydrin. Methanol is added for the total dissolution of the arabinose.

Comment 3: Details for characterization techniques should be provided such as temperature, conditions, gas flow rate, heating rate, etc.

Answer : Dear reviewer, we think that you raised a very good question. Details for characterization techniques have been added (see manuscript).

Comment 4: Proper references for the absorption bands at 2973, 2400, 1738, 1432, 1396, and 1366 cm-1 should be cited in the main text.

Answer : Dear reviewer, some references have been added (see manuscript).

Comment 5: Only meaningful values in the NMR graph should be displayed and discussed.

Answer : Dear reviewer, thank you very much for your comment. NMR graph has been revised.

Comment 6: What is the driving force or mechanism for the epoxy prepolymer investigated in this study acting as an effective corrosion inhibitor for mild steel?

Answer : Dear reviewer, in this study, we investigate the Langmuir adsorption mechanism model (see manuscript: 3.7 Adsorption study part).

Comment 7: Plenty of typos and grammar errors:

enables the to cooperate??

All used chemical products used in this -> Remove "used"

different carbons -> carbon atoms.

prepolymer was used and investigate -> investigated

a important -> an

epoxy prepolymer, which resulting -> Remove "which"

Answer : Dear reviewer, the plenty of typos and grammar errors has been revised and improved (see manuscript).

The authors would like to thank you for your time and effort. Thanks to these constructive comments,
